# Manifold Contrastive Learning with Variational Lie Group Operators

**Kion Fallah**                                                                    *kion@gatech.edu*
**Alec Helbling**
**Kyle A. Johnsen**
**Christopher J. Rozell**
*ML@GT*
*Georgia Institute of Technology*
*Atlanta, GA 30332*

**Reviewed on OpenReview:** *https://openreview.net/forum?id=lVE1VeGQwg*

## Abstract

Self-supervised learning of deep neural networks has become a prevalent paradigm for learning representations that transfer to a variety of downstream tasks. Similar to proposed models of the ventral stream of biological vision, it is observed that these networks lead to a separation of category manifolds in the representations of the penultimate layer. Although this observation matches the manifold hypothesis of representation learning, current self-supervised approaches are limited in their ability to explicitly model this manifold. Indeed, current approaches often only apply a pre-specified set of augmentations for "positive pairs" during learning. In this work, we propose a contrastive learning approach that directly models the latent manifold using Lie group operators parameterized by coefficients with a sparsity-promoting prior. A variational distribution over these coefficients provides a generative model of the manifold, with samples which provide feature augmentations applicable both during contrastive training and downstream tasks. Additionally, learned coefficient distributions provide a quantification of which transformations are most likely at each point on the manifold while preserving identity. We demonstrate benefits in self-supervised benchmarks for image datasets, as well as a downstream semi-supervised task. In the former case, we demonstrate that the proposed methods can effectively apply manifold feature augmentations and improve learning both with and without a projection head. In the latter case, we demonstrate that feature augmentations sampled from learned Lie group operators can improve classification performance when using few labels [1].

## 1 Introduction

Deep learning systems have made remarkable progress in decision-making tasks by leveraging large-scale, unlabeled datasets to learn neural representations. These representations, such as the activations at the penultimate layer of a deep neural network (DNN), have demonstrated efficacy in object recognition tasks with limited labels (Chen et al., 2020b), as well as the capability to generalize performance to several downstream tasks (Chen et al., 2020a; Caron et al., 2021). To effectively learn, DNNs must extract salient features, such as the natural variations of a category of data, from high-dimensional inputs into low-dimensional latent representations. Among geometric descriptions of this desired property, manifold models offer a natural explanation to how data existing in high dimensions can be parameterized by fewer degrees of freedom.

This explicit use of manifold models to represent visual invariances is also central to hypotheses underlying the ability of biological visual systems to achieve performance that generalizes across multiple tasks. For

---

[1]Code available at *https://github.com/kfallah/manifold-contrastive*.

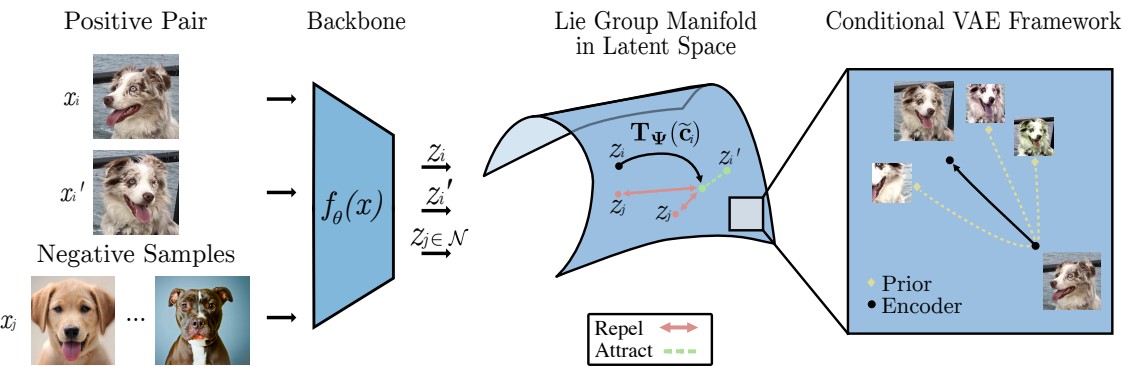

Figure 1: Proposed ManifoldCLR system for incorporating Lie group operators into the feature space of a contrastive learner. Given encoded features from a positive pair, manifold feature augmentations $\mathbf{T}_{\boldsymbol{\Psi}}(\widetilde{\mathbf{c}}_i)$ are applied before computing the contrastive objective. Here, $\mathbf{z}_j$ denote the set of negative pairs in the feature space. We apply the conditional VAE framework to learn coefficient distributions for manifold displacement between two points (encoder network $q_{\boldsymbol{\phi}}(\mathbf{c} \mid \mathbf{z}_i, \mathbf{z}_i')$) and identity-preserving augmentations from a single point (prior network $p_{\boldsymbol{\theta}}(\mathbf{c} \mid \mathbf{z}_i)$).

example, the ventral stream of the primate primary visual cortex has been suggested to produce linearly separable category manifolds which are robust to pose change (DiCarlo & Cox, 2007; DiCarlo et al., 2012; Majaj et al., 2015). Unfortunately, the computational mechanisms underlying this performance are not understood, and a large gap still exists between such systems and their deep learning counterparts. Indeed, deep learning methods are often expected to implicitly learn invariances from large-scale datasets (Moskalev et al., 2022), but often fail in distinguishing task-relevant and irrelevant changes to the input (Jacobsen et al., 2019). In the context of contrastive learning, part of this gap may be stated as the lack of explicit manifold modeling, limiting the number of unique views of an instance the model sees during training. Rather than contrast potential views from some general manifold structure in the dataset, these methods often rely on a set of pre-specified augmentations in the data space (Chen et al., 2020a; Cubuk et al., 2020).

In this work, we take a step towards closing this gap by proposing ManifoldCLR, a system for incorporating manifold structure describing natural data variations within the latent representations of a DNN while concurrently enriching such representations by sampling novel views along the manifold as feature augmentations during training. The proposed manifold model is parameterized by coefficients with a sparsity-promoting prior (Culpepper & Olshausen, 2009; Olshausen & Field, 1996), with learned distributions that can find both manifold paths between a pair of points, as well as identity-preserving feature augmentations from a single point. To make such sampling possible, we propose the variational Lie group operator (VLGO) model built upon recent advances in a variational sparse coding (Barello et al., 2018; Tonolini et al., 2020; Fallah & Rozell, 2022). With added block-diagonal structure on the learned operators, we are able to reduce memory usage of the model with minimal impact to downstream performance. We demonstrate the efficacy of our approach on self-supervised and semi-supervised benchmarks with image datasets (Krizhevsky, 2009; Coates et al., 2011; Deng et al., 2009).

## 2 Background

### 2.1 Manifold Learning

Manifold models have gained interest in the machine learning community for their geometric description of how data collected in high ambient dimension (e.g., pixels in an image) can vary with a few degrees of freedom. In the context of representation learning, this idea is formalized by the manifold hypothesis (Fefferman et al., 2016), which suggests that instances from a category lie near a low-dimensional manifold,

with low probability regions separating categories of data (Bengio et al., 2013). This hypothesis has been validated on many real-world datasets of interest (Pope et al., 2021), with theoretical implications on the sample complexity of learning underlying data distributions (Narayanan & Mitter, 2010). As such, manifolds have been incorporated into learned representations to disentangle factors of variation (Bouchacourt et al., 2021; Connor et al., 2023), provide equivariance to model outputs (Cohen & Welling, 2016), or give additional context when learning with limited labels (Belkin & Niyogi, 2002; Belkin et al., 2005). Many classical methods seek non-linear embeddings of data (Saul et al.; Tenenbaum, 2000; Belkin & Niyogi, 2001) or tangent plane approximations (Bengio & Monperrus, 2004; Rifai et al., 2011), but both are limited in their ability to interpolate or generate new points on the manifold.

In this work, we build upon a class of methods which provide a generative model of the manifold (Rao & Ruderman, 1998; Miao & Rao, 2007; Culpepper & Olshausen, 2009; Sohl-Dickstein et al., 2017) by learning the basis of a Lie algebra (Hall, 2015). Once passed through the matrix exponential, a point in this algebra provides a linear representation of a generator of a Lie group, denoting a displacement along a manifold. As an example, let $\mathcal{T}$ be a family of operators that forms a generic group. This means that (1) there exists a function $f : \mathcal{T} \times \mathcal{T} \to \mathcal{T}$ that maps pairs of transformations to another transformation such that $f$ is associative, (2) there exists a unique identity transformation $\exists t \in \mathcal{T} \; t(x) = x$, and (3) each transformation $t \in \mathcal{T}$ is invertible (Rao & Ruderman, 1998). Furthermore, restricting $\mathcal{T}$ to form a Lie group means that it applies continuous transformations parameterized by a vector of real numbers. In the context of this work, this parameterization comes from inferred coefficients that are used to take a linear combination of the learned operators. Although Lie groups form a subset of all differentiable manifolds, they have a long history of use as models of images (Rao & Ruderman, 1998) and visual perception (Dodwell, 1983). For a more comprehensive background, we refer readers to Hall (2015).

In this work, we describe this basis as a dictionary of $M$ operators $\mathbf{\Psi}_m \in \mathbb{R}^{D \times D}$. Given a point on the manifold $\mathbf{x}_i \in \mathbb{R}^D$, the Lie group operators model an infinitesimal transformation as a weighted sum of the learned dictionary

$$\dot{\mathbf{x}}_i = \left( \sum_{m=1}^{M} \mathbf{\Psi}_m c_m \right) \mathbf{x}_i, \tag{1}$$

where $\mathbf{c} \in \mathbb{R}^M$ is modeled with a sparse, factorial prior $p(\mathbf{c}) = \prod_{m}^{M} p(c_m)$ (Culpepper & Olshausen, 2009; Olshausen & Field, 1996). Given a batch of point pairs sampled nearby on the manifold $(\mathbf{x}_i, \mathbf{x}_i')$, the dictionary of operators can be learned through an alternating minimization scheme where coefficients $\mathbf{c}$ are first inferred with fixed operators (e.g., using a sparse inference procedure (Olshausen & Field, 1996)), followed by a gradient step with fixed coefficients to minimize the loss with respect to the operators (Culpepper & Olshausen, 2009):

$$\mathcal{L}_m(\mathbf{x}_i, \mathbf{x}_i', \mathbf{c}) = \| \mathbf{x}_i' - \mathbf{T}_{\mathbf{\Psi}}(\mathbf{c}) \mathbf{x}_i \|_2^2, \tag{2}$$

$$\mathbf{T}_{\mathbf{\Psi}}(\mathbf{c}) = \mathrm{expm} \left( \sum_{m=1}^{M} \mathbf{\Psi}_m c_m \right). \tag{3}$$

Learning these operators directly in the data space is often impractical due to dimensionality. As such, later work has extended this model to learning the operators after applying non-linear dimensionality reduction, $\mathbf{z}_i = f_{\boldsymbol{\theta}}(\mathbf{x}_i) \in \mathbb{R}^d$, where $d \ll D$ (Connor & Rozell, 2020; Connor et al., 2021; Bouchacourt et al., 2021; Connor et al., 2023; Cosentino et al., 2022a). Note that in this setting, the operators describe transformations between latent points and have dimensionality $\mathbb{R}^{d \times d}$. Most of these systems rely on an auto-encoding objective to prevent collapse to a trivial manifold (e.g., projecting every input data to a constant value allows one to learn all operators as identity), but are often limited by the fidelity of the decoder network in their ability to apply manifold transformations.

Furthermore, performing exact sparse inference $\mathbf{c}$ at each training step becomes overly prohibitive as dimensionality $d$ increases. Although fast first-order optimization algorithms have been proposed (Beck & Teboulle, 2009; Connor et al., 2023), such methods are not practical in deep learning systems that require upwards of hundreds of thousands of training iterations. Hence, alternative inference methods are necessary

when incorporating the operators into high-dimensional feature spaces (e.g., the penultimate layer of a ResNet backbone (He et al., 2016)).

## 2.2 Variational Inference

When dealing with probabilistic models with computationally infeasible inference procedures, variational inference has emerged as an effective approach (Zhang et al., 2018). For example, let $p_{\boldsymbol{\Psi}}(\mathbf{x}_i)$ be some arbitrary likelihood of input $\mathbf{x}_i$ parameterized by $\boldsymbol{\Psi}$. When maximizing this likelihood, marginalized over a latent variable $\mathbf{c}$, one may apply the evidence lower bound (ELBO) (Jordan et al., 1998) with a learned variational posterior $q_{\boldsymbol{\phi}}(\mathbf{c} \mid \mathbf{x}_i)$:

$$
\begin{aligned}
\log p_{\boldsymbol{\Psi}}(\mathbf{x}_i) &= \log \mathbb{E}_{p(\mathbf{c})}\left[p_{\boldsymbol{\Psi}}(\mathbf{x}_i \mid \mathbf{c})\right] \\
&= \log \mathbb{E}_{p(\mathbf{c})}\left[\frac{q_{\boldsymbol{\phi}}(\mathbf{c} \mid \mathbf{x}_i)}{q_{\boldsymbol{\phi}}(\mathbf{c} \mid \mathbf{x}_i)} p_{\boldsymbol{\Psi}}(\mathbf{x}_i \mid \mathbf{c})\right] \\
&\geq \mathbb{E}_{q_{\boldsymbol{\phi}}}\left[\log p_{\boldsymbol{\Psi}}(\mathbf{x}_i \mid \mathbf{c})\right] - D_{KL}\left(q_{\boldsymbol{\phi}}(\mathbf{c} \mid \mathbf{x}_i) \,\|\, p(\mathbf{c})\right).
\end{aligned}
$$

In practice, maximizing this lower bound is often presented as minimizing the negative lower bound. The rest of this work will use this presentation.

Recent methods have learned this variational posterior with a DNN employing the "reparameterization trick" (Kingma & Welling, 2014; Rezende et al., 2014), allowing one to simplify inference procedures as a single forward pass. Variational sparse coding approaches extend this procedure by encouraging sparsity in the latent variables $\mathbf{c}$ sampled from the variational posterior network. These approaches either use sparsity promoting priors (Barello et al., 2018), approximations to discrete variables (Tonolini et al., 2020), or straight-through estimations of soft-thresholding (Fallah & Rozell, 2022) to achieve the desired sparsity. In each of these previous works, inference is performed from a single input $\mathbf{x}_i$, with a Gaussian likelihood computed either using a linear dictionary or DNN decoder. In this work, we are interested in applying variational sparse coding to the Lie group operator model, using pairs of points for inference.

## 2.3 Self-Supervised Learning

Self-supervised learning (SSL) has emerged as an effective paradigm of leveraging unlabeled data to learn deep representations that transfer to several downstream tasks (He et al., 2020), such as learning with limited labels (Chen et al., 2020b). For example, contrastive learning (Hadsell et al., 2006) learns representations by encouraging similarity between representations of "positive pair", while encouraging dissimilarity between representations of "negative pairs". Different approaches have been proposed for selecting negative pairs, such as nearest neighbor instances (Dwibedi et al., 2021), memory banks of features (He et al., 2020; Chen et al., 2020c), or other instances within a batch (Chen et al., 2020a). Besides contrastive methods, negative-free approaches have emerged that either apply regularization to learned features (Ermolov et al., 2021; Bardes et al., 2022) or utilize teacher networks/stop-grad operations to prevent collapse to a trivial point (Grill et al., 2020; Chen & He, 2021; Caron et al., 2021).

To implement the contrastive learning objective, the InfoNCE loss (Oord et al., 2019) is often employed. First, DNN backbone feature pairs $(\mathbf{z}_i, \mathbf{z}_i') = (f_{\boldsymbol{\theta}}(\mathbf{x}_i), f_{\boldsymbol{\theta}}(\mathbf{x}_i'))$ corresponding to two views of an image instance $(\mathbf{x}_i, \mathbf{x}_i')$ are found by randomly sampling from a set of image space augmentations (Chen et al., 2020a; Sohn et al., 2020). Then, the temperature normalized InfoNCE objective is computed (Chen et al., 2020a):

$$
\mathcal{L}_{ctt}(\mathbf{z}_i, \mathbf{z}_i') = -\log \frac{\exp\left(-\|h_{\boldsymbol{\theta}}(\mathbf{z}_i) - h_{\boldsymbol{\theta}}(\mathbf{z}_i')\|_2^2/\tau\right)}{\sum_{j \in \mathcal{N}} \exp\left(-\|h_{\boldsymbol{\theta}}(\mathbf{z}_i) - h_{\boldsymbol{\theta}}(\mathbf{z}_j)\|_2^2/\tau\right) + \exp\left(-\|h_{\boldsymbol{\theta}}(\mathbf{z}_i) - h_{\boldsymbol{\theta}}(\mathbf{z}_i')\|_2^2/\tau\right)}, \tag{4}
$$

where $\mathcal{N}$ denotes the set of negative pairs and $\tau$ is a temperature hyper-parameter. Here, $h_{\boldsymbol{\theta}}$ is an additional normalized, non-linear projection from the backbone features. Recent work has argued that this projection learns a noisy estimate of the data manifold, encouraging backbone invariance to non-salient image features

---

**Algorithm 1** Variational Sparse Coding

---

**Input:** Input positive pair $\mathbf{z}_i$ and $\mathbf{z}'_i$, whether to use a SoftThreshold, threshold hyper-parameter $\zeta$, number of samples $J$.

$(\boldsymbol{\mu}_i, \mathbf{b}_i) \leftarrow g_{\boldsymbol{\phi}}(\text{sg}\,[\mathbf{z}_i \oplus \mathbf{z}'_i])$

**for** $j = 1$ **to** $J$ **do**

    $\boldsymbol{\epsilon}_i^j \sim U(-\frac{1}{2}, \frac{1}{2})$

    $\mathbf{s}_i^j \leftarrow \boldsymbol{\mu}_i + \mathbf{b}_i \circ \text{sign}(\boldsymbol{\epsilon}_i^j) \ln\left(1 - 2 \mid \boldsymbol{\epsilon}_i^j \mid\right)$

    **if** SoftThreshold **then**

        $\mathbf{c}_i^j \leftarrow \mathbf{s}_i^j + \text{sg}\left[\mathcal{T}_\zeta\left(\mathbf{s}_i^j\right) - \mathbf{s}_i^j\right]$

    **else**

        $\mathbf{c}_i^j \leftarrow \mathbf{s}_i^j$

    **end if**

    $\mathcal{L}_m^j \leftarrow \mathcal{L}_m(\mathbf{z}_i, \mathbf{z}'_i, \mathbf{c}_i^j)$

**end for**

$\mathbf{c}_i \leftarrow \arg\min_j \mathcal{L}_m^j$

---

(Cosentino et al., 2022b). Oddly, this non-linear projection is often discarded after training, with backbone features $\mathbf{z}_i$ used instead for downstream tasks.

## 2.4 Related Work

Other work has considered unsupervised feature learning using manifold models. The sparse manifold transform (SMT) applies slow feature analysis after a linear sparse coding step to obtain a generative manifold embedding (Chen et al., 2018), with extensions to learning unsupervised "white-box" representations (Chen et al., 2023). In this work, rather than applying linear sparse coding to a single point, we infer coefficients to find non-linear manifold paths between a pair of points. Applying slow feature analysis to the sparse coefficients of the Lie group operators is an interesting direction that we leave to future work. Likewise, other work has considered manifold SSL by encouraging maximum manifold capacity under an elliptical manifold model (Yerxa et al., 2023). The model proposed in this work has the advantage of learning an arbitrary manifold structure coinciding with a Lie group.

In a similar work, Ibrahim et al. (2022) apply Lie group operators for robustness to unseen views in SSL. However, our work differs in two key ways: (1) we propose a variational framework for coefficients rather than a deterministic estimate, allowing one to sample coefficients, and (2) we incorporate feature augmentations from Lie group operators into contrastive learning, providing significant improvements in performance and allowing us to quantify coefficient distributions which preserve identity. Both of these changes are critical for the goal of this work, which is to incorporate manifold feature augmentations into the contrastive learning process. Indeed, a deterministic encoder precludes the ability to sample coefficients, with the encoder in Ibrahim et al. (2022) requiring information regarding the transformation applied between the point pair as input (see Equation 4 of Ibrahim et al. (2022)), making it impossible to get coefficients from a single input.

# 3 Methods

## 3.1 Variational Lie Group Operator Model

To effectively learn and apply the Lie group operators, one needs a method to perform quick coefficient inference at each training step. The quality of the learned operators is completely dictated by how effective inferred coefficients transport from an initial to a target point on the manifold. Furthermore, not every manifold augmentation can be applied with the same magnitude at every point on the manifold. This means that incorporating feature augmentations requires knowledge of which operators can be applied for a given input point and the extent to which they can be applied. To address these requirements for quick inference and learned coefficient distributions from which one can sample, we build upon advances in variational

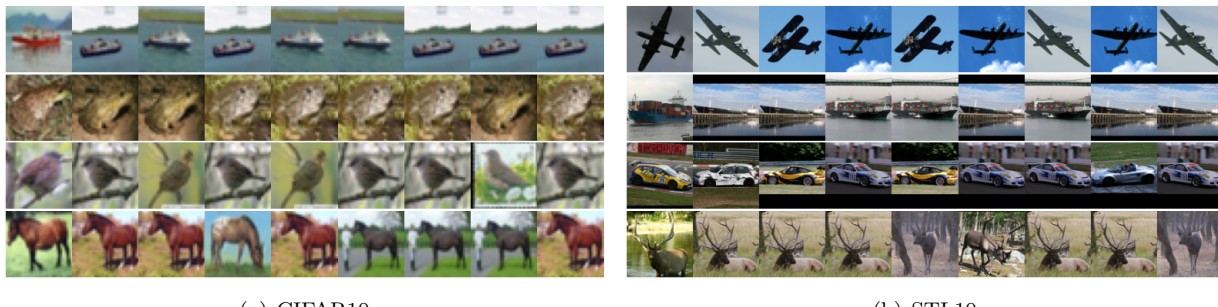

(a) CIFAR10                                              (b) STL10

Figure 2: Nearest neighbor visualization of feature augmentations sampled from the learned prior. For each row, the first column denotes the input to the prior. Each subsequent column is found by taking a random sample from the prior, applying an operator augmentation, and visualizing the image corresponding to the nearest feature in the dataset. It can be qualitatively seen that the prior network learns coefficient statistics that result in augmentations that preserve class-identity.

sparse coding (Fallah & Rozell, 2022; Tonolini et al., 2020; Barello et al., 2018). Given features close on the manifold $(\mathbf{z}_i, \mathbf{z}'_i)$, we sample sparse coefficients by reparameterizing distribution parameters encoded by a recognition network $\mathbf{c} \sim q_\phi(\mathbf{c} \mid \mathbf{z}_i, \mathbf{z}'_i)$[1] (Kingma & Welling, 2014; Rezende et al., 2014). We summarize the reparameterization procedure in Algorithm 1, with the option to incorporate soft-thresholding and take $J$ best-of-many samples for effective machine-precision sparsity (Fallah & Rozell, 2022; Bhattacharyya et al., 2018).

After applying the variational lower bound, we obtain the following variational Lie group operator objective, which includes a weighted KL divergence term (Higgins et al., 2017):

$$\mathcal{L}_m\left(\mathbf{z}_i, \mathbf{z}'_i, \mathbf{c}\right) + \beta \mathcal{L}_{kl}\left(\mathbf{z}_i, \mathbf{z}'_i, \mathbf{c}\right) \tag{5}$$

In this framework, one can either fix prior distribution parameters $p_\theta(\mathbf{c} \mid \mathbf{z}_i)$ as in standard variational inference, or encode them with a separate DNN as in the conditional variational autoencoder (CVAE) framework (Sohn et al., 2015). During training, samples are drawn from the encoder network to concurrently train the operators $\boldsymbol{\Psi}_m$ with encoder weights $\phi$ (and backbone/prior weights $\theta$ when applicable).

Like the CVAE (Sohn et al., 2015), the encoder $q_\phi$ and prior $p_\theta$ encode distributions to solve different tasks. The encoder outputs coefficient statistics for reconstruction of $\mathbf{z}'_i$, analogous to class labels in the CVAE framework. On the other hand, the prior network solves a predictive task in encoding statistics to generate other points on the manifold from an initial $\mathbf{z}_i$. Learning the prior network provides several advantages. First, it provides flexibility for the encoder network to deviate from fixed statistics when certain operators are more or less likely to be used for an initial point $\mathbf{z}_i$. Second, it allows one to sample feature augmentations along the manifold to apply both as a synthetic positive pair on the feature manifold during SSL pretraining, or as strong feature augmentation during tasks like semi-supervised learning (Kuo et al., 2020; Sohn et al., 2020). In Section 4.2, we demonstrate that learning the prior leads to better identity-preservation than naively sampling from a fixed prior.

## 3.2 ManifoldCLR: Manifold Contrastive Learning

Given the variational Lie group operator framework, one can learn a manifold describing the natural variations present in a dataset within the representations of a DNN (while also regularizing such representations to match the data manifold). This is done by sampling coefficients from the encoder network $\mathbf{c} \sim q_\phi(\mathbf{c} \mid \mathbf{z}_i, \mathbf{z}'_i)$ to minimize the manifold loss $\mathcal{L}_m$. Furthermore, one can directly sample transformations along the manifold to enrich the number of views that are applied during instance discrimination. To do this, one may sample

---

[1]Features are detached, concatenated, and fed through a multi-layer perceptron followed by a linear projection for each distribution parameter.

coefficients $\widetilde{\mathbf{c}} \sim p_{\boldsymbol{\theta}}(\mathbf{c} \mid \mathbf{z}_i)$ from the learned prior network, apply manifold feature augmentations to obtain a synthetic view $\widetilde{\mathbf{z}}_i = \mathbf{T}_{\boldsymbol{\Psi}}(\widetilde{\mathbf{c}})$, and contrast the positive pair $(\widetilde{\mathbf{z}}_i, \mathbf{z}'_i)$.

This allows one to both enrich the novel views of an instance, as well as learn coefficient statistics in the prior network that preserve identity. We hypothesize that this latter outcome is a result of negative samples in the contrastive loss, which encourages prior augmentations to remain on the object manifold, visualized on the right in Figure 1. We note that previous approaches for learning identity-preserving manifold transformations have required labels (Connor et al., 2023), whereas the methods proposed here are entirely unsupervised.

Finally, the resultant learning objective can be written as:

$$\min_{\boldsymbol{\Psi}, \boldsymbol{\theta}, \boldsymbol{\phi}} \mathcal{L}_{ctt}(\widetilde{\mathbf{z}}_i, \mathbf{z}'_i) + \lambda \mathcal{L}_m(\mathbf{z}_i, \mathbf{z}'_i, \mathbf{c}) + \beta \mathcal{L}_{kl}(\mathbf{z}_i, \mathbf{z}'_i, \mathbf{c}) \tag{6}$$

where $\boldsymbol{\Psi}$ are manifold operator parameters, $\boldsymbol{\theta}$ are DNN backbone and prior network parameters, $\boldsymbol{\phi}$ are coefficient network parameters, and $\lambda$ and $\beta$ are loss scaling hyper-parameters. All terms and parameters are learned concurrently, end-to-end from random weight initialization. We note that this is a simplified training procedure in comparison to previous works with Lie group operators that use multiple training phases (Connor & Rozell, 2020; Connor et al., 2023) or pretrained backbone models (Connor et al., 2023; Ibrahim et al., 2022).

### 3.2.1 Modified Manifold Objective

Learning the Lie group operators can be impractical in cases where $d$ is large due to memory constraints in computing the matrix exponential. To solve this, we propose additional block-diagonal constraints onto the learned operators. We do this by breaking up the features of our point pair into segments of length $b$. Rather than constrain the operators to be block-diagonal at each training step, they are initialized as $d/b$ separate dictionaries $\boldsymbol{\Psi}^j$ of shape $\mathbb{R}^{b \times b}$. After inferring coefficients for each pair of features, we compute the objective for each segment separately[3]:

$$\mathcal{L}_m(\mathbf{z}, \mathbf{z}', \mathbf{c}) = \sum_{j=0}^{d/b-1} \left\| \operatorname{sg}\Big(\mathbf{z}'[jb : (j+1)b]\Big) - \mathbf{T}_{\boldsymbol{\Psi}^j}(\mathbf{c})\mathbf{z}[jb : (j+1)b] \right\|_2^2. \tag{7}$$

Furthermore, we find that including the stop-grad operation, written sg and indicating no gradient computed through the term, on $\mathbf{z}'$ helps prevent feature collapse. This technique has been previously applied in systems such as SimSiam (Chen & He, 2021). Under this approach, the number of parameters needed to learn a dictionary of 128 operators in a 512 dimensional ResNet-18 (He et al., 2016) backbone is reduced from **33.6M** parameters to **4.2M** parameters. This leads to a significant reduction in memory use during training with minimal impact to downstream performance. We ablate the impact of these choices in Section 4.2.3.

## 4 Experiments

### 4.1 Synthetic Dataset

To measure the effectiveness and speed-up of the VLGO model, we first evaluate on a synthetic dataset, comparing to the training procedure proposed in Culpepper & Olshausen (2009). We train on 5000 points generated on a Swiss roll manifold, using the fast iterative shrinkage-thresholding (FISTA) algorithm (Beck & Teboulle, 2009) for exact sparse inference as a baseline. At each iteration, we select 500 points and randomly sample the point pair from the 20th to 60th nearest neighbors. In total, we train for 1000 epochs and measure three key metrics, (1) the mean-squared error between transported point pairs $\mathcal{L}_m$, (2) the cumulative runtime, and (3) the $\ell_1$ loss of inferred coefficients to measure sparsity. For hyper-parameters, we set $M = 6$, the weight of the $\ell_1$ penalty as 0.6, and the weight of the F-norm penalty on the operators as 1e−3.

We compare to the VLGO using both a standard Laplacian prior (Barello et al., 2018) and thresholded samples from a Laplacian (Fallah & Rozell, 2022), evaluating at different sample counts $J$. We use the same

---

[3]We use $[a : b]$ as in array notation to denote the subset of dimensions from $a$ to $b$ from the feature vector.

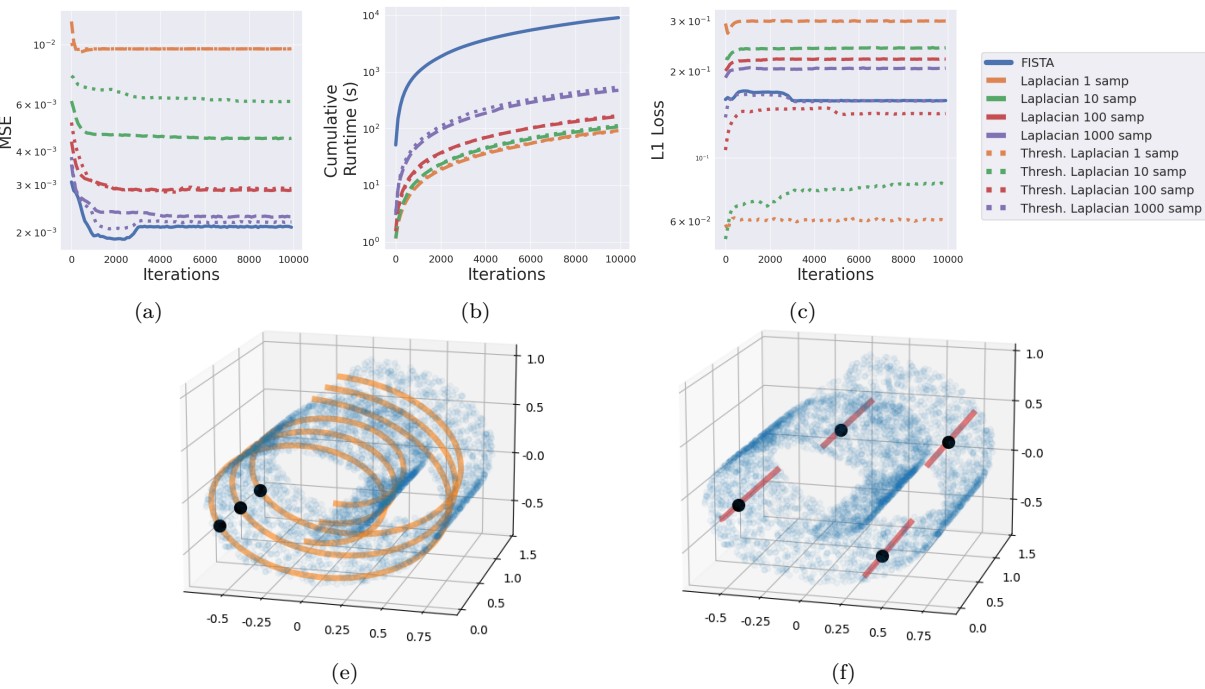

Figure 3: Comparison of Lie group operators trained with exact and variational sparse inference on synthetic Swiss roll dataset. For each method, we plot (a) mean-squared error, (b) cumulative runtime, and (c) $\ell_1$ penalty of coefficients. (e,f) Extrapolated paths of the two operators learned by VLGO. Black dots indicate starting point $\mathbf{x}_i$, with subsequent path found by applying Lie group operators $\mathbf{T}_{\mathbf{\Psi}}(\mathbf{c}_i)\mathbf{x}_i$ with coefficients from a range with fixed step size $\mathbf{c}_i = [-N_c, \ldots, N_c]$.

hyper-parameters as the FISTA baseline with a fixed prior and $\beta = 5e-3$. The results are shown in Figure 3, where three main conclusions can be drawn: (1) as the number of samples drawn from the encoder increases, variational methods match FISTA in MSE, (2) when using more samples, thresholding gives the same MSE as a standard Laplacian with much lower $\ell_1$ loss, (3) variational methods are approximately 15x faster in training time than FISTA. We note that FISTA and the thresholded Laplacian result in two non-zero operators after training, whereas the standard Laplacian results in three non-zero operators. We attribute this to exact sparsity, allowing weight decay to shrink unused operators in the FISTA and thresholded Laplacian models.

## 4.2 Image Datasets

### 4.2.1 Experimental Setup

Next, we compare the performance of the ManifoldCLR system equipped with VLGO to standard contrastive learning, as well as other baselines for training without a projection head or incorporating feature augmentations into the learning process. For the main results, we train each method using the SimCLR framework for selecting positive and negative pairs (Chen et al., 2020a), only adding additional feature augmentations on our positive pair when relevant. Specifically, let $(t_i, t'_i) \sim \mathcal{T}$ be two augmentations sampled from those presented in SimCLR (described in Appendix 6.2.1). For each input image in a batch $\mathbf{x}_i$, we use $(\mathbf{x}_i, \mathbf{x}'_i) = (t_i(\mathbf{x}_i), t'_i(\mathbf{x}_i))$ as the input point pair for each evaluated method. Although we use this framework to measure the relative benefits of proposed methods, we note that they are compatible with other contrastive learning systems that use the InfoNCE loss (Chen et al., 2020c). For example, we compare to NNCLR (Dwibedi et al., 2021) by selecting nearest neighbors as our point pairs in Table 2. We leave methods for incorporating manifold feature augmentations into other self-supervised frameworks to future work.

We evaluate both using the standard linear readout protocol (Chen et al., 2020a), shown in Table 1, and a modified semi-supervised objective, shown in Figure 4. We test on a variety of datasets, including CIFAR10 (Krizhevsky, 2009), STL10 (Coates et al., 2011), and TinyImageNet (Deng et al., 2009). We build on the experimental setup of Ermolov et al. (2021), training a ResNet-18 (He et al., 2016) with a batch size of 512 for 1000 epochs across all experiments. For methods that use a projection head, we normalize the projected features before computing the InfoNCE loss. When training without a projection head, we find that using MSE in the InfoNCE provides superior performance (ablated in Table 7). We train all methods with a single NVIDIA A100 GPU, with an approximate runtime of 9 hours for baselines and 24 hours for ManifoldCLR on TinyImageNet. Additional analysis on the computational and memory complexity can be found in Appendix 6.1.1. Full details on the experimental setup and evaluation is included in Appendix 6.2.1. Before discussing the results, we introduce each of the baselines for feature augmentations.

**ManifoldMixupCLR** uses the common linear interpolation strategy for feature augmentation proposed in Manifold Mixup (Verma et al., 2019a). For each positive and negative pair, a random interpolation constant is sampled $\lambda_i \sim U(0,1)$, and applied to form a feature augmentation $\widetilde{\mathbf{z}}_i = \lambda_i \mathbf{z}_i + (1 - \lambda_i)\mathbf{z}_i'$.

**ManifoldCLR** uses the VLGO model to sample coefficients from a learned prior network $\widetilde{\mathbf{c}} \sim p_{\boldsymbol{\theta}}(\mathbf{c} \mid \mathbf{z}_i)$ that is used to apply manifold feature augmentations $\widetilde{\mathbf{z}}_i = \mathbf{T}_{\boldsymbol{\Psi}}(\widetilde{\mathbf{c}})\mathbf{z}_i$. This is done concurrently with learning the Lie group operators by minimizing $\mathcal{L}_m$ with coefficients from the encoder network. We set $\beta = 1e{-}5$ for each method and $\lambda = 10$ and $\lambda = 1$ with and without a projection head, respectively. Furthermore, we set the dictionary size $M = (16, 64, 128)$ for datasets (CIFAR10, STL10, TinyImageNet). Finally, we set the block size of our operators $b = 64$ for each experiment.

**DirectCLR** (Jing et al., 2022) contrasts only the first $b$ components of the backbone features when training without a projection head. We apply it as a baseline method that improves upon contrastive learning without a projection head.

**ManifoldDirectCLR** combines ManifoldCLR and DirectCLR by applying the manifold loss only to the first $b = 64$ feature components. Furthermore, augmentations and contrasting is only performed on these first few feature components.

Table 1 shows that when trained with a projection head, ManifoldCLR provides an improvement in all datasets over SimCLR. Without a projection head, ManifoldCLR performs the best among feature augmentation methods. Performance benefits over ManifoldMixupCLR indicate the importance in identifying non-linear paths in the latent space for feature augmentations. Notably, we find that the benefits of ManifoldCLR and DirectCLR are complementary and that combining the two gives a higher performance than SimCLR with a projection head. We find this result satisfying, since it affirms the purpose of the projection head as a means to estimate the data manifold (Cosentino et al., 2022b). ManifoldCLR is capable of estimating this manifold structure directly in the backbone with VLGO, removing redundant model weights from the training process while also providing a generative model of the manifold.

To test the applicability of the ManifoldCLR framework to different strategies for selecting point pairs, we also compare to NNCLR (Dwibedi et al., 2021) in Table 2. For this experiment, we follow a methodology similar to Table 5 in Dwibedi et al. (2021), where we contrast nearest neighbor point pairs with only random cropping applied as the augmentations (i.e., no color jitter or grayscale). We present our results with a projection head on CIFAR10 taken over three random trials. Although these experiments have lower separability than the SimCLR augmentations, this is to be expected, based on the results presented in Table 5 in Dwibedi et al. (2021). We deviate from NNCLR in a few ways for these experiments. First, to increase the stability of training, we follow the methodology of Connor et al. (2023) and pre-compute all the point pairs using an auxiliary feature space (DINO model (Caron et al., 2020) pretrained on ImageNet). We note that this leads to a significant benefit (approximately +40%) for both the NNCLR baseline and ManifoldCLR. Furthermore, for simplicity, we do not include the additional prediction head used in NNCLR. This, however, has very minimal impact (approximately 0.4%), as shown in Table 7f in Dwibedi et al. (2021).

Table 1: Linear probe accuracy on various image datasets with different methods for contrasting point pairs with our without a projection head. Results averaged over three trials. We underline the best performing method(s) overall and **bold** the best method(s) without a projection head. See Section 4.2 for details on methods used.

| Method | Projection Head | CIFAR10 | STL-10 | TinyImageNet |
|---|:---:|---|---|---|
| SimCLR-MLP | ✓ | 89.04% | 86.68% | 39.74% |
| SimCLR-Linear | ✓ | 87.78% | 85.18% | 36.88% |
| ManifoldCLR-MLP | ✓ | 90.03% | 87.33% | 42.77% |
| SimCLR-None | ✗ | 88.58% | 84.53% | 36.26% |
| ManifoldMixupCLR | ✗ | 86.52% | 83.02% | 38.33% |
| ManifoldCLR | ✗ | 88.89% | 84.99% | 38.68% |
| DirectCLR | ✗ | 88.46% | 84.46% | 36.20% |
| ManifoldDirectCLR | ✗ | **89.73%** | **87.47%** | **42.22%** |

Table 2: Linear probe accuracy compared between NNCLR and ManifoldCLR using nearest neighbor point pairs. The images are only augmented with a random crop, as described in Appendix 6.2.1. Both approaches use an MLP projection head. Results averaged over three random trials.

| Method | CIFAR10 |
|---|---|
| NNCLR-MLP | $66.13 \pm 0.20\%$ |
| ManifoldCLR-MLP | $\mathbf{69.43 \pm 0.17\%}$ |

#### 4.2.2 Semi-supervised Learning with Manifold Feature Augmentations

To highlight the benefits of generating manifold feature augmentations for down-stream tasks, we evaluate on a semi-supervised task using five labels per class and consistency regularization (Sohn et al., 2020) in Figure 4. For each dataset, we freeze the weights of the best performing backbone network from Table 1 and train a single hidden-layer MLP with different methods for feature augmentations. We train under this protocol since fine-tuning the backbone weights would result in a deviation from the manifold learned during contrastive pre-training. We have had preliminary success in fine-tuning while including the ManifoldCLR objective (hence adapting the manifold to the classification task), but we leave such experiments for future work. Here, we aim to demonstrate that the VLGO can provide feature augmentations that improve classification performance with limited labels. Given labeled features $(\mathbf{z}_i^l, y^l)$ and unlabeled features $\mathbf{z}_i^u$, we train a classifier $r_{\boldsymbol{\theta}}$ using the following loss:

$$\min_{\boldsymbol{\theta}} \frac{1}{B^l} \sum_{i=1}^{B^l} H\left(\mathbf{q}_i^l, y_i^l\right) + \frac{1}{N^u} \sum_{b=1}^{B^u} \mathbb{1}\left[\mathbf{q}_b^u \geq \tau\right] H\left(\widetilde{\mathbf{q}}_b^u, \widehat{y}_b^u\right).$$

Here, $B^l$ and $B^u$ are the batch size of the labeled and unlabeled dataset, respectively. $\mathbf{q}_i^l = r_{\boldsymbol{\theta}}(\mathbf{z}_i^l)$ and $\mathbf{q}_b^u = r_{\boldsymbol{\theta}}(\mathbf{z}_b^u)$ are the normalized logits from the classifier corresponding to class probabilities for the labeled and unlabeled features, respectively. $N^u = \sum_{b=1}^{B^u} \mathbb{1}\left[\mathbf{q}_b^u \geq \tau\right]$ is the number of unlabeled examples with confidence above $\tau$ and $H(\cdot)$ is the cross-entropy loss. We follow the methodology of Sohn et al. (2020) and use the class predictions of unlabeled data of which the classifier is sufficiently confident as pseudo-labels $\widehat{y}_b^u = \arg\max \mathbf{q}_b^u$ for predictions on augmented examples $\widetilde{\mathbf{q}}_b^u = r_{\boldsymbol{\theta}}(\widetilde{\mathbf{z}}_b^u)$. To get augmentations, we apply Lie group transformations $\widetilde{\mathbf{z}}_b^u = \mathbf{T}_{\boldsymbol{\Psi}}(\widetilde{\mathbf{c}}_b)\mathbf{z}_b^u$ using coefficients sampled from the learned prior $\widetilde{\mathbf{c}}_b \sim p_{\boldsymbol{\theta}}(\mathbf{c} \mid \mathbf{z}_b^u)$. We focus our comparison to other methods for incorporating feature augmentations.

We train each dataset and method with $B^l = 32$, $B^u = 480$ randomly sampled examples at each iteration, using a single hidden-layer MLP with 2048 hidden units. We train with the AdamW optimizer (Loshchilov & Hutter, 2019) with a fixed learning rate and weight decay of 5e−4. For CIFAR10 and STL10, we set $\tau = 0.95$. For TinyImageNet, we set $\tau = 0.7$. Furthermore, we evaluate using an exponential moving average (EMA) of

Table 3: Average percent improvement in semi-supervised accuracy over baseline over 50 splits with varying datasets and methods for incorporating feature augmentations using 5 labels/class. Best method(s) for each dataset are bolded.

| Method | CIFAR10 | STL-10 | TinyImageNet |
|---|---|---|---|
| Pseudo-labeling | $3.99 \pm 1.88\%$ | $\mathbf{0.34 \pm 4.90}\%$ | $\mathbf{1.25 \pm 0.28}\%$ |
| Manifold Mixup ICT | $-1.46 \pm 0.48\%$ | $-3.26 \pm 1.11\%$ | $-0.01 \pm 0.11\%$ |
| FeatMatch | $\mathbf{5.08 \pm 1.20}\%$ | $1.35 \pm 0.50\%$ | $0.26 \pm 0.15\%$ |
| Variational Lie Group Operators | $\mathbf{8.61 \pm 2.57}\%$ | $\mathbf{5.47 \pm 2.37}\%$ | $\mathbf{1.58 \pm 0.30}\%$ |

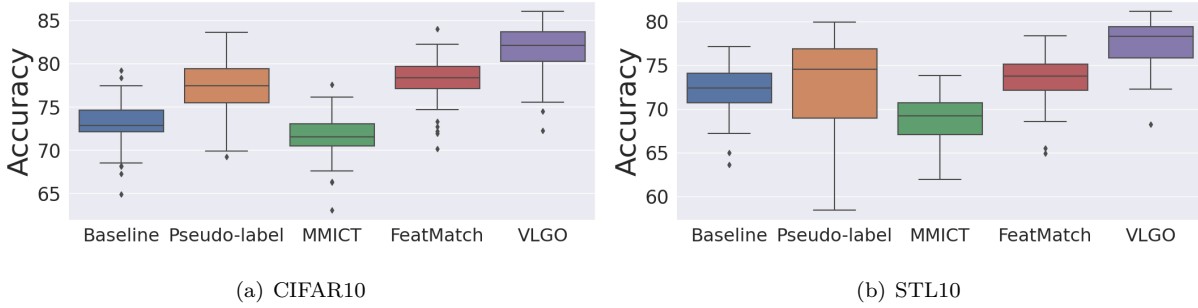

(a) CIFAR10        (b) STL10

Figure 4: Semi-supervised experiments using a frozen backbone and learned, single hidden layer MLP. Comparison methods leverage unlabeled data to encourage consistency in classifier output after applying feature augmentations. Taken over 50 data splits used for each method.

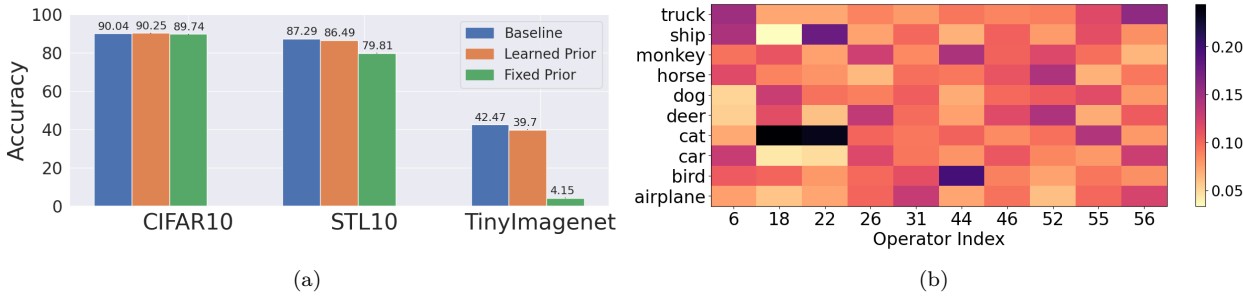

(a)        (b)

Figure 5: (a) Identity-preservation of prior measured by linear classifier accuracy on frozen features after sampling augmentations. Standard error is below $0.21\%$ for each method. (b) Average, normalized coefficients per class sampled from learned prior on STL10 dataset. Coefficients can be used to analyze which transformations are more or less likely for different input points and classes.

our classifier weights at each training step with weight $0.999$. We train each method for $5,000$ iterations, except FeatMatch (Kuo et al., 2020) which we train for $10,000$ iterations to give the attention head sufficient iterations to learn meaningful augmentations.

To account for high variance in trial performance (e.g., certain splits of five labels per class are more challenging than others), we present the result with respect to percent improvement over a baseline that only uses a standard cross-entropy loss on labeled features in Table 3. We note that VLGO provides the highest accuracy across almost every data split (plotted in Figure 7). Furthermore, we include paired t-tests in Table 9, showing that accuracy improvements from the Lie group augmentations are statistically significant at the group level. We compare to pseudo-labeling without augmentations (Lee, 2013), FeatMatch (Kuo et al., 2020), and Manifold Mixup interpolation consistency training (MMICT) (Verma et al., 2019b). For FeatMatch, we use all the labeled data as the prototypes for the attention-based feature augmentation module with four

Table 4: Linear probe accuracy and manifold fit on TinyImageNet without a projection head ablating various components from the ManifoldCLR system. Results averaged over three trials. See Section 4.2.3 for details on each system.

| System | Stop-grad on $\mathbf{z}'$ | Lie Group Augmentations | Manifold Loss $\mathcal{L}_m$ | Learned Prior | Linear Probe Acc |
|--------|------------|-------------|------------|--------|------------|
| $\mathcal{S}_0$ | ✓ | ✓ | ✓ | ✓ | **38.68** |
| $\mathcal{S}_1$ | ✗ | ✓ | ✓ | ✓ | 31.97 |
| $\mathcal{S}_2$ | ✓ | ✗ | ✓ | ✓ | 37.37 |
| $\mathcal{S}_3$ | ✓ | ✓ | ✗ | ✓ | 36.57 |
| $\mathcal{S}_4$ | ✓ | ✓ | ✓ | ✗ | 36.85 |

attention heads. For MMICT, we apply a random linear interpolation sampled from $U(0, 1)$ between two randomly sampled unlabeled points and apply an MSE loss between the mixup of logits predicted by the EMA model and the logits of the interpolated features. We note that this differs from the methodology of Verma et al. (2019b) in that we interpolate features rather than the input data. This method is meant to compare the benefit of learning non-linear paths with the Lie group operators. Among these methods, it can be seen that VLGO provides the most benefit in Figure 4. We include experiments using more labels in Appendix 6.2.2, where the trial variance is reduced while the relative benefit of VLGO is also minimized.

To better understand the types of augmentations encoded by the prior, we visualize the nearest neighbor image to several feature augmentations in Figure 2. To quantify the degree of identity-preservation, we classify augmented features from our validation set under a linear probe classifier in Figure 5(a). Here, it can be seen that a learned prior has significantly superior identity-preservation over a fixed prior, with the gap widening as the dataset complexity increases. Furthermore, by analyzing the normalized encoded coefficients, averaged per class, from the prior network in Figure 5(b), one can draw insight on the magnitude of transformations that can be applied per class, as well as which transformations are shared between classes. For example, operator 52 is most likely for 'horse' and 'deer', operator 55 for 'dog' and 'cat', and operator 56 for 'truck', 'car', and 'airplane'. We emphasize that previous work with Lie group operators required all class labels to learn identity-preserving coefficient distributions (Connor et al., 2023), as opposed to the unsupervised approach in this work.

### 4.2.3 Ablations

Next, we ablate several components in the proposed system ($\mathcal{S}_0$) and measure the impact to linear readout accuracy on TinyImageNet trained without a projection head in Table 4. First, we observe the impact of removing the stop-grad operation in Equation 7 as $\mathcal{S}_1$. Next, we test removing the prior augmentations from the contrastive loss in Equation 4 and removing the manifold loss (i.e., $\lambda = 0$ in Equation 6) in systems $\mathcal{S}_2$ and $\mathcal{S}_3$, respectively. We note similar degradation of performance when using coefficients sampled from the encoder network as augmentations in the contrastive loss. Finally, we measure the impact of replacing the learned prior with a fixed prior for contrastive loss augmentations in $\mathcal{S}_4$.

To evaluate the effect of hyper-parameters on the ManifoldCLR system, we measure linear probe accuracy while varying different hyper-parameters on the TinyImageNet dataset with a projection head in Figure 6. First, it can be seen that dictionary size follows an inverted U-shape, with too few dictionary entries causing underfitting and too many causing overfitting. We find that this is the hyper-parameter that is most critical to tune. On the other hand, we see that changing the weight $\lambda$ of the manifold loss $\mathcal{L}_m$ provides little difference on downstream performance. Lastly, we see that imposing a block diagonal constraint on operators has little impact on linear probe performance above a block size of 64 (with significant benefits in computational and memory costs). As one may expect, however, we find that reducing the block size leads to a reduction of the effective rank (Roy & Vetterli, 2007) of backbone features. We report these values in Table 8.

## 5    Discussion & Limitations

In this work, we demonstrate that incorporating novel views via manifold feature augmentations improves the performance of contrastive learning across several image datasets. To learn the manifold structure of data in a way that is amicable to sampling, we propose VLGO, a variational framework for the Lie group operator model. This allows one to quickly infer coefficients describing the manifold displacement between a pair of points, while also learning identity-preserving transformations to be applied to a single point in the feature space. We demonstrate that this model can be used in contrastive learning to improve performance with and without a projection head, potentially obviating the need for a projection head completely. After contrastive pre-training, we also demonstrate that feature augmentations from the prior can be used for improvement in semi-supervised learning.

One limitation of the proposed model is memory usage from the lack of fp16 support in current implementations of the matrix exponential (Bader et al., 2019; Paszke et al., 2019). Here, we address this constraint by imposing block-diagonal structure in the learned Lie group operators. Development of new numerical methods for the matrix exponential is future work that can dramatically improve the applicability of VLGO to larger scale models. Furthermore, although our work focuses on the benefits in contrastive learning, interesting future directions may consider evaluation in other self-supervised frameworks.

### Acknowledgements

The authors would like to thank Brighton Ancelin and Coleman DeLude for their helpful discussions about Lie groups. This work was partially supported by NSF CAREER award CCF-1350954 and ONR grant number N00014-15-1-2619.

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

# 6 Appendix

## 6.1 Appendix A: Training Variational Lie Group Operators

When training the VLGO, there are several metrics of success we use to evaluate the model. For a given point pair $(\mathbf{x}_i, \mathbf{x}_i')$ and coefficients $\mathbf{c}_i$, the most important metric is the distance improvement (DI) from the operators:

$$\text{DI} = \frac{\|\mathbf{x}_i' - \mathbf{T}_\Psi(\mathbf{c})\mathbf{x}_i\|_2^2}{\|\mathbf{x}_i' - \mathbf{x}_i\|_2^2}. \tag{8}$$

This metric indicates how effective the operators/coefficient inference strategy is in transporting from $\mathbf{x}_i$ to $\mathbf{x}_i'$. When training the VLGO in the feature space of a model, this metric is especially important to ensure that the model is not minimizing $\mathcal{L}_m$ by trivially collapsing the features. It is important to also measure the average magnitude of non-zero coefficients and the average Frobenius norm of the Lie group operators to ensure they are not increasing without bound. To help ensure stability, we apply gradient clipping to both the Lie group operators and the weights of the coefficient encoder and prior at each training step. Additionally, we clamp the estimated distribution parameters from the encoder and prior network. It is critical to not set the clipping/clamping values too high, as it can harm performance.

It is also important to learn operators that are approximately stable, with almost entirely imaginary eigenvalues (i.e., real components near zero) defining cyclic paths. This is important to prevent applied augmentations from causing an unbounded growth in the magnitude of the features (see appendix of Connor et al. (2023)). We have found that the initialization of the Lie group operators affect the spectra of the final learned operators. As such, we initialize each operator in the dictionary with the following block-diagonal form:

$$\begin{bmatrix} \alpha_1^m & \beta_1^m & \dots & 0 & 0 \\ -\beta_1^m & \alpha_1^m & \dots & 0 & 0 \\ 0 & 0 & \ddots & 0 & 0 \\ \vdots & \vdots & 0 & \alpha_{d/2}^m & \beta_{d/2}^m \\ 0 & \dots & 0 & -\beta_{d/2}^m & \alpha_{d/2}^m \end{bmatrix},$$

where $\alpha_i^m$ and $\beta_i^m$ denote the real and imaginary component of the $i$th and $i+1$th eigenvalue, respectively. Note that imaginary eigenvalues come in conjugate pairs. For the ManifoldCLR experiments, we set $\alpha_i^m = 1.0e{-}4$ and $\beta_i^m = 6.0$.

Finally, in the case when we learn the prior, we apply a warmup of the estimated prior distribution parameters. For the first 5000 iterations of training, we apply a linear warm-up from the encoded prior parameters and fixed prior parameters. Let $(\mu_i, b_i) = g_\theta(\mathbf{z}_i)$ be prior Laplacian parameters for input $\mathbf{z}_i$ and $\kappa \in [0, 1]$ be a warm-up scalar. In the first 5000 iterations, we set the prior parameters as:

$$(\mu_i, b_i) = \Big(\kappa\mu_i + (1 - \kappa)\mu_0, \kappa b_i + (1 - \kappa)b_0\Big),$$

where $\mu_0$ and $b_0$ are initial, fixed prior parameters. We use initial prior distribution parameters in our evaluation in Figure 5(a).

### 6.1.1 Computational and Memory Complexity of Lie Group Operators

In this work, we take three steps to improve computational complexity: (1) variational inference to avoid the need for an iterative optimization procedure at each training step, (2) block diagonal structure on the learned operators, and (3) reducing the magnitude of the coefficients via the scale parameter of the prior distribution or additional $\ell_2$ regularization. For the matrix exponential, we use an optimized Taylor series polynomial (Bader et al., 2019) with computational complexity $O(KN^3)$, where $K$ is a term that depends on the norm of the matrix, as described in Equation 7 of Bader et al. (2019). Hence, reducing the magnitude of the coefficients leads to a decrease in matrix norm in Equation 3 of our manuscript, and hence the computational complexity.

Table 5: Comparison of memory usage and average iteration time between SimCLR and ManifoldCLR on TinyImageNet with a batch size of 512 using an Nvidia A100 GPU

| Method | GPU Memory (GB) | Avg Time per Iter (sec) |
|---|---|---|
| SimCLR | 13.4 | 0.0902 |
| ManifoldCLR | 15.2 | 0.4284 |

Table 6: Comparing the effect of block dimension of ManifoldCLR on memory usage and average iteration time on TinyImageNet with a batch size of 512 using four Nvidia RTX 6000 GPUs

| Block Dimension | GPU Memory (GB) | Avg Time per Iter (sec) |
|---|---|---|
| 32 | 25.80 | 0.4735 |
| 256 | 52.42 | 0.5279 |
| 512 | 70.68 | 0.9051 |

We include an additional experiment comparing the empirical computational and memory complexity of different methods on the TinyImagenet dataset. From Table 5, it can be seen that ManifoldCLR incurs only a slight memory overhead compared to SimCLR, with the previously stated limitation of a higher computational complexity. This comes with the advantage of learned operators that can be applied for downstream tasks. Fortunately, Table 6 suggests that imposing a block diagonal constraint significantly reduces computational complexity. We caution against comparison between Table 5 and Table 6 since they use different GPU architectures and Table 6 requires storing the model on each GPU.

## 6.2 Appendix B: Image Experiments

### 6.2.1 Experimental Setup

For each dataset, we train a ResNet-18 (He et al., 2016) with the AdamW optimizer (Loshchilov & Hutter, 2019) for 1000 epochs using a batch size of 512. We set the backbone and projection head learning rate to $3.0e-3$ for CIFAR10 and $2.0e-3$ for STL10 and TinyImageNet. For every model, we set the learning rate of the Lie group operators and coefficient encoder to $1.0e-3$ and $1.0e-4$, respectively. We set the weight decay equal to $1.0e-5$ for the backbone, projection head, and coefficient encoder and equal to $1.0e-3$ for the Lie group operators. We use a cosine annealing scheduler with a 10 warm-up epochs and a minimum learning rate of $1.0e-5$ for all parameters.

For all datasets, we modify the first convolutional layer of the ResNet-18 to have a 3x3 kernel size with a stride and padding of 1 and no bias term. For CIFAR10, we also remove the max pool layer. We use images of size 32x32 for CIFAR10 and 64x64 for STL10 and TinyImageNet. When applicable, we use a single hidden layer projection head with 1024 hidden units, BatchNorm1d, and an output dimension of 128. To augment the images, we first apply a random color jitter with brightness, contrast, and saturation set to 0.4 and hue set to 0.1 with probability 0.8. We then apply a random grayscale with probability 0.1. Afterwards, we apply a random resize/crop with scale between 0.2 and 1.0 and ratio between 0.75 and 1.33. Finally, we apply a random horizontal flip with probability 0.5 and normalize the image using the standard mean and standard deviation value for each dataset.

For our coefficient encoder architecture, we use a two-hidden-layer MLP with a leaky ReLU non-linearity. The encoder takes concatenated, detached features as input (dimension equal to 1024) with hidden layers with 2048 units. The output of the MLP is 512, after which a single linear layer is used to estimate log scale and shift for each coefficient component. Our prior network uses a similar architecture, except with an input dimension 512 and the second hidden layer of the MLP with 1024 units. Due to memory constraints from taking multiple samples, we use a standard Laplacian prior for all ManifoldCLR experiments except CIFAR10 with a projection head. In that case, we apply a threshold $\zeta = 0.01$ with $J = 20$ samples. For all methods, we set the initial prior shift equal to 0.05 and scale equal to 0.01.

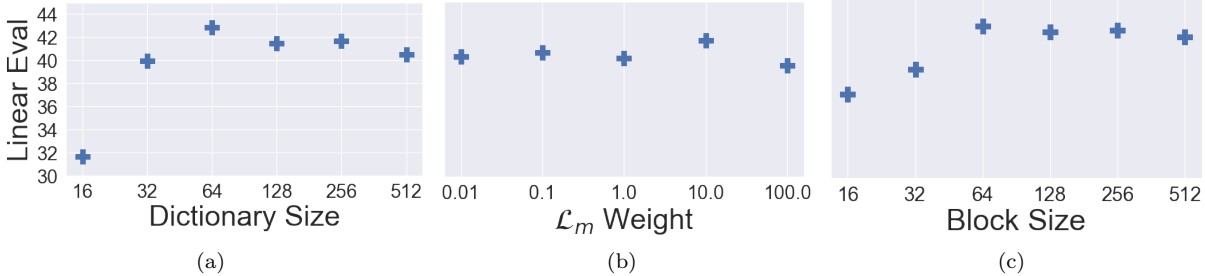

Figure 6: Linear probe accuracy on a TinyImageNet model trained with a projection head while varying different components of the ManifoldCLR system. These include the (a) dictionary size $M$, (b) weight on the manifold loss term $\lambda$, and (c) block size $b$ for the block-diagonal constraint on operators.

Table 7: Linear probe accuracy for DirectCLR (Jing et al., 2022) and DirectCLR+ManifoldCLR (named ManifoldDirectCLR). Evaluated using an InfoNCE loss with normalized features (i.e., cosine similarity) and without normalized features (i.e., MSE). All methods trained without a projection head. Best method(s) for each dataset are bolded.

| Method | Normalized | CIFAR10 | STL-10 | TinyImageNet |
|---|---|---|---|---|
| SimCLR-None | ✓ | 80.49% | 78.29% | 24.47% |
| DirectCLR | ✓ | 85.55% | 82.99% | 33.29% |
| SimCLR-None | ✗ | 88.58% | 84.53% | 36.26% |
| DirectCLR | ✗ | 88.46% | 84.46% | 36.20% |
| ManifoldCLR | ✗ | 88.89% | 84.99% | 38.68% |
| ManifoldDirectCLR | ✗ | **89.73%** | **87.47%** | **42.22%** |

To perform the linear probe evaluation, we freeze the backbone and encode features of the entire dataset without augmentations. We then train a single linear layer for 500 epochs using the Adam optimizer. We use an exponential decay on our learning rate at the end of each epoch, starting at $1.0e{-2}$ and ending at $1.0e{-5}$.

### 6.2.2 Additional Results

Table 8: Comparison of effective rank of backbone features between SimCLR and ManifoldCLR with different block sizes in the Lie group operators. All methods trained on TinyImageNet with a projection head. Block size of 512 indicates no block-diagonal structure in the operators.

| Method | Block Size | Effective Rank |
|---|---|---|
| SimCLR | – | 6.01 |
| ManifoldCLR | 512 | 5.87 |
| ManifoldCLR | 256 | 5.75 |
| ManifoldCLR | 128 | 5.37 |
| ManifoldCLR | 64 | 5.25 |
| ManifoldCLR | 32 | 4.23 |
| ManifoldCLR | 16 | 3.32 |

Table 9: Paired t-test p-values between variational Lie group operators and every other method over 50 random data splits on semi-supervised experiment using **5 labels/class** from main text. Alternative hypothesis is that mean of VLGO accuracy is greater than each method.

| Method | CIFAR10 | STL-10 | TinyImageNet |
|---|---|---|---|
| Baseline | $1.33e{-}28$ | $1.29e{-}21$ | $7.40e{-}38$ |
| Pseudo-labeling | $3.58e{-}16$ | $5.85e{-}10$ | $6.10e{-}13$ |
| Manifold Mixup ICT | $4.91e{-}32$ | $2.48e{-}27$ | $9.63e{-}40$ |
| FeatMatch | $1.25e{-}14$ | $2.42e{-}18$ | $3.48e{-}35$ |

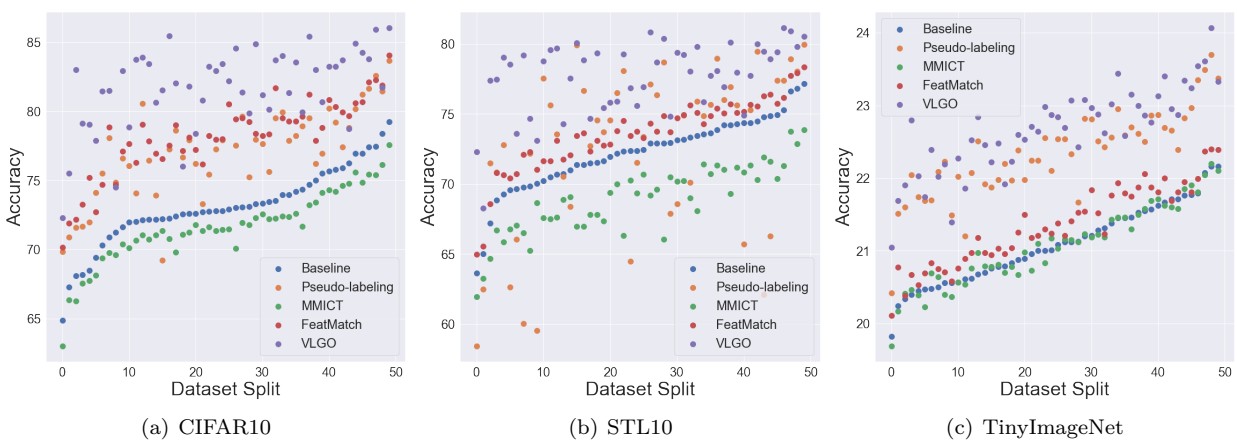

Figure 7: Per trial performance, sorted by baseline performance, of semi-supervised experiments with **5 labels/class** taken over 50 data splits. For almost every split, VLGO provides the highest accuracy.

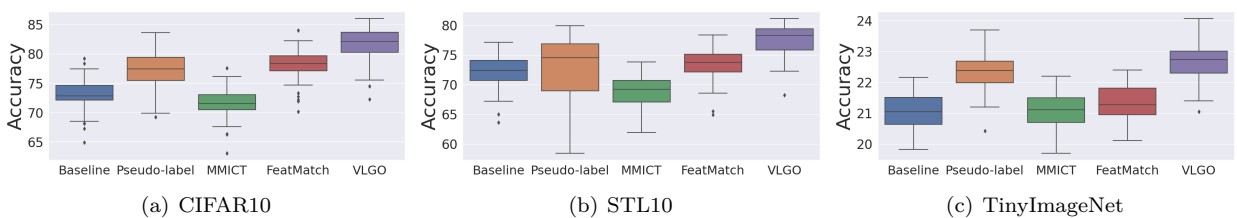

Figure 8: Semi-supervised experiments using a frozen backbone and learned, single hidden layer MLP with **5 labels/class**, taken over 50 data splits. Best viewed zoomed in.

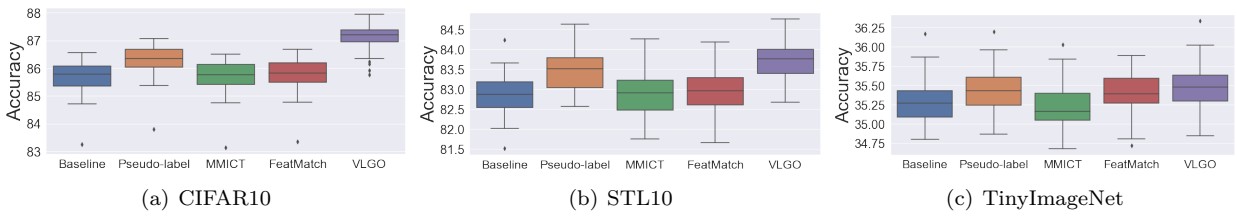

Figure 9: Semi-supervised experiments using a frozen backbone and learned, single hidden layer MLP with **50 labels/class**, taken over 50 data splits. Best viewed zoomed in.

Table 10: Average percent improvement in semi-supervised accuracy over baseline over 50 splits with varying datasets and methods for incorporating feature augmentations using **50 labels/class**. Best method(s) for each dataset are bolded.

| Method | CIFAR10 | STL-10 | TinyImageNet |
|---|---|---|---|
| Pseudo-labeling | $0.59 \pm 0.12\%$ | $\mathbf{0.62 \pm 0.21}\%$ | $\mathbf{0.16 \pm 0.11}\%$ |
| Manifold Mixup ICT | $0.02 \pm 0.07\%$ | $0.02 \pm 0.09\%$ | $-0.06 \pm 0.21\%$ |
| FeatMatch | $0.10 \pm 0.08\%$ | $0.08 \pm 0.08\%$ | $0.10 \pm 0.21\%$ |
| Variational Lie Group Operators | $\mathbf{1.42 \pm 0.37}\%$ | $\mathbf{0.85 \pm 0.30}\%$ | $\mathbf{0.22 \pm 0.12}\%$ |

Table 11: Average percent improvement in semi-supervised accuracy over baseline over 50 splits with varying datasets and methods for incorporating feature augmentations using **100 labels/class**. Best method(s) for each dataset are bolded.

| Method | CIFAR10 | STL-10 | TinyImageNet |
|---|---|---|---|
| Pseudo-labeling | $0.30 \pm 0.09\%$ | $\mathbf{0.31 \pm 0.13}\%$ | $0.11 \pm 0.09\%$ |
| Manifold Mixup ICT | $0.02 \pm 0.08\%$ | $0.02 \pm 0.08\%$ | $0.00 \pm 0.22\%$ |
| FeatMatch | $0.01 \pm 0.07\%$ | $0.02 \pm 0.10\%$ | $\mathbf{0.66 \pm 0.25}\%$ |
| Variational Lie Group Operators | $\mathbf{0.82 \pm 0.19}\%$ | $\mathbf{0.46 \pm 0.19}\%$ | $0.14 \pm 0.10\%$ |

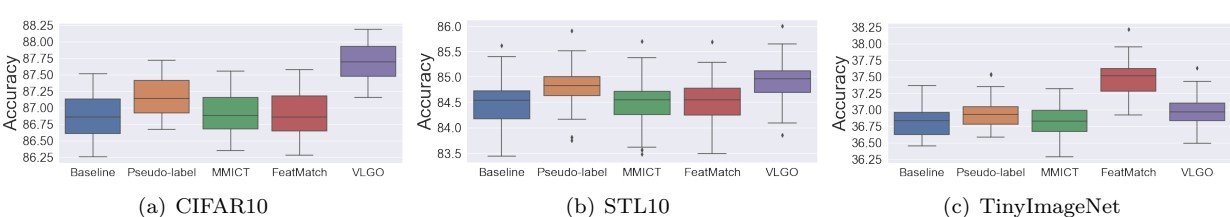

(a) CIFAR10  (b) STL10  (c) TinyImageNet

Figure 10: Semi-supervised experiments using a frozen backbone and learned, single hidden layer MLP with **100 labels/class**, taken over 50 data splits. Best viewed zoomed in.

