# OpenReview forum: "Manifold Contrastive Learning with Variational Lie Group Operators"
_TMLR — Accepted by TMLR_

### Review · Reviewer_FZ1K · 2023-09-12

**Summary Of Contributions:**

Self-supervised learning has become a prevalent paradigm for learning general purpose representations from unlabeled data. However, most of existing methods rely on pre-defined augmentations and do not explicitly enforce manifold structure in the learned representations. This is suboptimal, as manifold models provide a natural framework for capturing invariant features, similar to how biological vision systems are believed to create linearly separable category manifolds.

To address these limitations, this paper proposes ManifoldCLR - a principled approach to incorporate manifold modeling into contrastive self-supervised learning. The main contribution is a variational Lie group operator (VLGO) model that captures manifold variations and allows sampling of augmentations along the learned manifold. The model is integrated into contrastive learning, where manifold samples provide additional positive pairs to improve representation learning. Experiments demonstrate benefits of ManifoldCLR on self-supervised benchmarks and downstream tasks compared to baselines.

**Audience:**

Yes

**Broader Impact Concerns:**

I have No Broader Impact concerns.

**Claims And Evidence:**

Yes

**Requested Changes:**

**Discussion and major concerns:**
- I would recommend adding analysis of the computational and memory complexity of the ManifoldCLR framework, either in the main paper or an appendix. Understanding the operational costs is important for practitioners looking to adopt this approach.
- While the learned augmentations do seem to preserve identity well, an open question remains around whether they provide sufficiently challenging positive samples for contrastive learning. The pre-defined augmentations in standard contrastive learning can sometimes generate very challenging positive pairs, forcing the model to learn more robust and generalizable features. The identity-preserving priors in ManifoldCLR may not distort the data to the same extent. Further analysis or visualization of the augmentations could provide insight into the level of difficulty and study the impact on generalization performance.
- The infoNCE loss presented in eq(4) differs slightly from the loss defined in other works, such as CPC, SimCLR etc, where they compute the similarity score by taking the inner product of vectors. In eq (4), the square of the $\ell_2$ norm is calculated instead. The empirical results of ablation Table 4 shows that without normalized feature outperformed normalized features. This is very interesting. I would greatly appreciate it if the author could provide more insights into the motivation behind and advantages of this modification.
- In Algorithm 1, line 2, what is $g_\theta$ stands for? Is it a typo, and should it be referring to "$q_\theta$"?
- I am wondering would the effectiveness of the prior augmentations diminish when labels are available during downstream tasks?

**Strengths And Weaknesses:**

**Strengths:**
- Proposed a principled way to incorporated manifold modeling into self-supervised learning representations. The method is end-to-end trainable and doesn't require a pre-trained feature extractor, unlike some prior works applying manifold operators.
- Quantitatively demonstrated their framework, MainfoldCLR, improves performance on self-supervised benchmarks over baseline methods.
- VLGO provides a flexible way to model manifold variations and sample useful augmentations for downstream tasks like semi-supervised learning.
- The learned prior distribution provides a way to quantify which augmentations are identity-preserving at different points on the manifold. This is learned in a completely unsupervised way.

**Weakness:**
- Due to the high computation and memory cost of matrix exponential, it is challenging to scale up this method to larger datasets. Even though the author proposed block diagonal modification to alleviate this issue, the ManifoldCLR spent 3X runtime than baselines on the TinyImageNet dataset, implying the computational cost is still expensive.
- While effective manifold augmentations are demonstrated for images, it's unclear if this approach would work well for other data modality like audio, video, graphs, etc. Additionally, having experimental results on larger datasets would enhance the generalizability of the conclusions.
- If it is available, providing intuitions for what semantic transformations the learned operators represent.

---

> ### Author Response · Authors · 2023-11-05
> **Response to Reviewer FZ1K**
>
> We thank the reviewer for their time and thoughtful feedback. We agree with the reviewer that ManifoldCLR provides a “principled way to incorporate manifold modeling into self-supervised learning” with performance improvements that are “quantitatively demonstrated”. Furthermore, we agree with the reviewer that more analysis of the complexity of the ManifoldCLR, especially the matrix exponential, are warranted. We will revise the manuscript with discussion from the rebuttal written here.
>
> We would like to clarify to the reviewer that the InfoNCE loss in Eq. 4 reduces to the same objective as the noted works when the projected features are normalized, since the squared norm would equal the negative inner produce with an added constant. We note that other works in the literature use the same presentation of this loss (e.g., Eq. 2 of [1]). In our experiments, such as Table 4, we observe that using the squared L2 norm only has superior performance when we do not apply any normalization/projection. Additionally, we use the notation g_{\theta} to denote the deterministic “reparameterization trick” like previous works in variational inference (see Eq. 4 of [2]), as opposed to stochastic samples from q_{\phi}.
>
> In this work, we take three steps to improve computational complexity: (1) variational inference to avoid the need for an iterative optimization procedure at each training step, (2) block diagonal structure on the learned operators, and (3) reducing the magnitude of the coefficients via the scale parameter of the prior distribution or additional l2 regularization. For the matrix exponential, we use an optimized Taylor series polynomial [3] with computational complexity O(KN^3), where K is a term that depends on the norm of the matrix, as described in Eq. 7 of [3]. Hence, reducing the magnitude of the coefficients leads to a decrease in matrix norm in Eq. 3 of our manuscript, and hence the computational complexity.
>
> We include an additional experiment comparing the empirical computational and memory complexity of different methods on the TinyImagenet dataset. From Table 1, it can be seen that ManifoldCLR incurs only a slight memory overhead compared to SimCLR, with the previously stated limitation of a higher computational complexity. This comes with the advantage of learned operators that can be applied for downstream tasks. Fortunately, Table 2 suggests that imposing a block diagonal constraint significantly reduces computational complexity. We caution against comparison between Table 1 and 2 since they use different GPU architectures and Table 2 requires storing the model on each GPU.
>
> Although we agree with the reviewer that visualizations and a study of the semanticity of the learned augmentations would be interesting, we believe it would require developing new methodology and is out of the scope of this current work. Other recent work has demonstrated that Lie group operators learned in the feature space of auto-encoders learn semantic, generalizable augmentations [4]. Future work may also consider extending encoder inversion techniques to visualize augmentations [5]. Unfortunately, these methods rely on features from the middle layer of the network and are not applicable to augmentations from our operators which are applied in the final layer of the network. In our work, we rely on self-supervised metrics like in Table 1 and semi-supervised experiments in Table 2 to demonstrate that learned augmentations are sufficiently challenging and effective without and with labels, respectively.
>
> Table 1: Comparison of SimCLR and ManifoldCLR on TinyImageNet with a batch size of 512 using an Nvidia A100 GPU
> | Block Dimension | GPU Memory (GB) | Avg Time per Iter (sec) |
> |:---------------:|:---------------:|:-----------------------:|
> |      SimCLR     |       13.4      |          0.0902         |
> |   ManifoldCLR   |       15.2      |          0.4284         |
>
>
> Table 2: Comparison of block dimension of ManifoldCLR on TinyImageNet with a batch size of 512 using four Nvidia RTX 6000 GPUs
> | Block Dimension | GPU Memory (GB) | Avg Time per Iter (sec) |
> |:---------------:|:---------------:|:-----------------------:|
> |        32       |       25.8      |          0.4735         |
> |       256       |      52.42      |          0.5279         |
> |       512       |      70.68      |          0.9051         |
>
>
> [1] “Understanding Dimensional Collapse in Contrastive Self-supervised Learning,” Jing et al., 2021.
>
> [2] “Auto-Encoding Variational Bayes,” Kingma et Welling, 2013.
>
> [3] “Computing the Matrix Exponential with an Optimized Taylor Polynomial Approximation,” Bader et al., 2019.
>
> [4] “Learning Identity-Preserving Transformations on Data Manifolds,” Connor et al., 2023.
>
> [5] “What makes instance discrimination good for transfer learning?,” Zhao et al., 2020.

---

> > ### Comment · Reviewer_FZ1K · 2023-11-13
> >
> > Thanks for the authors' detailed response to my questions and additional experiment results. My major concerns have been addressed. Looking forward to reviewing the revised manuscript.

---

### Review · Reviewer_pMNS · 2023-10-20

**Summary Of Contributions:**

This paper introduces Lie group operators as a way to learn manifold structure in the context of contrastive learning. The operators are applied in a lower dimensional feature space and learnt using variational sparse coding techniques. The resulting training objective is a sum of the usual InfoNCE loss for contrastive learning with learnt transformations applied, the manifold reconstruction loss and the loss for the VAE used for variational sparse coding. A block-diagonal approximation is further applied to make learning the Lie operators tractable. The framework is evaluated in the context of standard self-supervised and semi-supervised settings where it is shown to outperform other contrastive learning and semi-supervised learning techniques.

**Audience:**

Yes

**Claims And Evidence:**

No

**Requested Changes:**

- Clarify how close-by examples on the manifold ($z_i$ and $z^{'}_i$) are selected
- Provide more background and detail on Lie groups and variational sparse coding
- Ablation studies to show the effectiveness of the variational approach for learning coefficients over a determinstic estimate (e.g. in Ibrahim et al. (2022))
- [Optional] Show that VLGO works with other self-supervised learning frameworks

**Strengths And Weaknesses:**

Strengths:
- Using Lie groups for manifold learning in the context of contrastive learning is interesting
- Experimental results are good, showing improvements over competing methods
- Ablation studies provided for many design choices
- Paper provides good discussion of related works

Weaknesses:
- It is unclear from the paper how the different views of the same example/close by examples on the manifold ($z_i$ and $z^{'}_i$) are selected, which is key to the algorithm. I would like to understand how this is done before making a decision on the soundness of the framework. For instance, if they are done using transformations then those should be representable by Lie groups for the framework to make sense.
- The paper is very similar to Ibrahim et al. (2022) as discussed in the related work, though there are 2 key differences highlighted. Ablation studies showcase the importance of incorporating feature augmentations from Lie group operators into contrastive learning, but there is no study showing the benefit of the variational approach used in this paper.
- The VLGO component seems to be generic but results are only shown with the SimCLR framework.
- On the one hand the manifold loss is shown to be important for performance in Table 3, but in a different set of ablation experiments on $\lambda$, the paper concludes that changing the weight "provides little difference on downstream performance". These two results should be somehow reconciled.
- The paper does not include sufficient background on several of the components to make it self-contained. In particular, an introduction to Lie groups/algebra would be helpful, as would a bit more detail on the variational sparse-coding component in the main text (e.g. what is the factorial prior used).

Other comments:
- In Figure 2, many of the augmented images appear very similar - are there actually differences (e.g. 2(a) first row 2, 4, 7, 8, 9 and 3, 5, 6)?
- Would be interesting to understand what the operators in Fig 5b are doing and if they make sense for the classes

Typos:
- Citation format should be corrected to use parenthetical citations unless referring to authors.
- (Page 3, just above Eq 1) "the Lie group operators model an infinitesimal transformation" - should this be "Lie algebra operators"

---

> ### Author Response · Authors · 2023-11-05
> **Rebuttal to Reviewer pMNS**
>
> We thank the reviewer for their time and thoughtful feedback, which we believe will strengthen our work. We are glad the reviewer finds manifold contrastive learning interesting and that the experimental results show “improvements over competing methods”. We are happy to revise our manuscript to fix our citation format and provide additional background on Lie groups and variational sparse coding, but will focus our current rebuttal on addressing technical concerns. To specifically address the question of our prior, we select a factorial Laplacian prior as in [1]. We agree with the reviewer that our claim about the manifold reconstruction loss in 4.2.3 was imprecise. Table 6 indeed suggests a degradation in performance as the weighting hyper-parameter is reduced below its optimal value, corroborating conclusions from Table 3. We meant to suggest that the model performance is less sensitive to this hyper-parameter in comparison to the dictionary size. We will refine our claim in our revision.
>
> In section 4.2.1, we state that we use the SimCLR framework for selecting our point pairs, meaning that we take the same instance undergoing two different transformations sampled from the augmentations presented in SimCLR (i.e., random cropping, color jitter, etc). We agree that this should be made more clear and will revise our manuscript to further clarify. We had preliminary success using the strategy applied in [2], taking two instances that are nearest neighbors in the feature space of a pre-trained self-supervised model, but pursued the current strategy for simplicity of presentation. In our revised manuscript, we will include a comparison between picking nearest neighbors and NNCLR [3]. We believe this will demonstrate both the generality of ManifoldCLR to the point pair and contrastive framework. Finally, we argue that the transformations need not necessarily correspond with a Lie group transformation in the data space, since prior evidence suggests that the intermediary layers of DNNs unravel and flatten data manifolds (potentially making Lie groups a strong inductive bias for other classes of transformations) [4].
>
> Although we initially considered a comparison to (Ibrahim et al. 2022) as the reviewer suggested, their work proposes a deterministic mapping over coefficients and provides no way to sample augmentations. Indeed, their encoder relies on side information about the transformation between a point pair and cannot encode coefficients from a single point (see Eq. 4 of their work). If the reviewer has a specific comparison experiment within the context of our work they had in mind, we would be happy to include it in our revision.
>
> With regard to Figure 2, we draw random samples from our prior and note that certain transformed samples may have the same nearest neighbor. Unfortunately, we have no decoder to reconstruct and visualize these transformations. However, we argue for their utility due to the increase in performance in Table 1 and Table 2. We agree that developing techniques to further understand the semanticity of these transformations is interesting (similar to work that investigates Lie groups in the feature space of auto-encoders [2]), but out of scope for the current work.
>
>
> [1] “Sparse coding with an overcomplete basis set: a strategy employed by V1?,” Olshausen et Field, 1997.
>
> [2] “Learning Identity-Preserving Transformations on Data Manifolds,” Connor et al., 2023.
>
> [3] “With a Little Help from My Friends: Nearest-Neighbor Contrastive Learning of Visual Representations,” Dwibedi et al., 2021.
>
> [4] “Separability and geometry of object manifolds in deep neural networks,” Cohen et al., 2020.

---

> > ### Comment · Reviewer_pMNS · 2023-11-26
> >
> > Thanks for the response - most of my concerns are addressed; I encourage the authors to further highlight the stated differences with (Ibrahim et al. 2022) in the manuscript.
> >
> > However, some items like the additional NNCLR experiments are not provided here; I would like to see the revised paper (the paper can be revised during this discussion phase) before being convinced that the paper should be accepted. As is, it requires a major revision to address all reviewers' concerns.

---

> > > ### Author Response · Authors · 2023-11-29
> > > **Response**
> > >
> > > Dear Reviewer pMNS,
> > >
> > > Thank you for your patience as we work to get the additional experiments and revisions completed. We have just posted results comparing to NNCLR. We hope this addresses remaining technical concerns. We will now focus our attention on addressing the presentation concerns of our work with the requested revisions to the manuscript. Please let us know if you have any additional concerns or doubts.
> > >
> > > Best,
> > > Authors

---

### Review · Reviewer_2HHn · 2023-10-29

**Summary Of Contributions:**

This paper investigates learning a self-supervised contrastive image model jointly with a model of local transformations along an image manifold. Local transformations are represented as a Lie group of matrix transformations that take a point on the image manifold as input and output a neighboring point on the image manifold. The local transformations are the weighted sum of a dictionary of learned matrices with coefficients that follow a sparse distribution. The motivation of using a Lie group generative model as part of the self-supervised learning pipeline is to enrich the contrastive embedding space with by modeling latent Lie group transformations. The contrastive model is trained using standard InfoNCE loss along with a variational loss that learns a CVAE modeling the Lie group transforms in the latent embedding space. The CVAE model consists of a learned prior that takes the starting state and learns an output that is reparameterized into a sparse distribution on weighting coefficients, along with an encoder network used for variational approximation that takes a pair of neighbors and outputs parameters for a variational weighting distribution. Samples from the latent Lie group samples are used as augmentations in addition to standard SimCLR augmentations. Experiments show that the applying the method along with SimCLR augmentations outperforms the base SimCLR model and other manifold variants of SimCLR in downstream linear probing tasks. Experiments using features from the model for weakly supervised classification show improved performance compared to baselines.

**Audience:**

Yes

**Broader Impact Concerns:**

Broader impacts are not discussed, but this does not impact my assessment of the work.

**Claims And Evidence:**

Yes

**Requested Changes:**

I do not have any significant requests for changes at this time.

**Strengths And Weaknesses:**

*Strengths*

* The proposed method is an interesting and relevant synthesis of self-supervised learning and generative model using Lie group transformations. There is a natural synergy between local manifold augmentations generated by Lie groups and those required by self-supervised learning and the direction seems promising.
* Empirical results show good performance compared to the base SimCLR model and other manifold-based contrastive learning approaches.
* The authors found ways to tackle computational limitations from the dimensionality of the embedding space by imposing block sparse structure on the learned transformation matrices.
* Results show a relatively small gap between models trained with and without a projection head, in contrast to SimCLR and other similar models. This could lead to more interpretable embedding spaces accompanied by models of local transformation structure.
* Each component was ablated on the non-trivial task of Tiny ImageNet linear probe accuracy.

*Weaknesses*

* The method still relies on the use of hand-chosen image augmentations. The learned augmentations in the latent space essentially re-learn hand-chosen image augmentations. Thus, the method does not achieve the ultimate goal of contrasting views from the image manifold rather than views from hand-crafted transformations as discussed in the introduction. Nonetheless, this approach could be a step towards such a method.

---

> ### Author Response · Authors · 2023-11-05
> **Response to Reviewer 2HHn**
>
> We thank the reviewer for taking the time to review our work and provide thoughtful feedback. We agree with the reviewer that the combination of manifold learning and self-supervised learning seems promising, and that ManifoldCLR takes steps to overcome existing computational limitations with superior performance to a base SimCLR model.
>
> To address concerns about learning hand-crafted augmentations, as well as those expressed by Reviewer pMNS, we plan to include an additional experiment comparing to contrasting nearest neighbor pairs in our revised manuscript (similar to experiments done in NNCLR [1]). We hope that this will also demonstrate the applicability of ManifoldCLR in other contrastive learning systems.
>
> [1] “With a Little Help from My Friends: Nearest-Neighbor Contrastive Learning of Visual Representations,” Dwibedi et al., 2021.

---

### Author Response · Authors · 2023-11-29
**NNCLR Results**

Dear reviewers,

We thank you for your patience as we work through the requested experiments and revisions in this busy time of the year. I am happy to share some promising results on our NNCLR experiments.

For these experiments, we follow a methodology similar to Table 5 of NNCLR [1], where we contrast nearest neighbor point pairs with only cropping applied as the augmentations (i.e., no color jitter or grayscale). We present our results with a projection head on CIFAR10 taken over three random trials. Although these experiments have lower separability than the SimCLR augmentations, this is to be expected compared to the results presented in Table 5 of NNCLR [1]. We deviate from NNCLR in a few ways for these experiments. First, to increase the stability of training, we follow the methodology of [2] and pre-compute all of the point pairs using an auxiliary feature space (DINO model pretrained on ImageNet). We note that this leads to a significant benefit (roughly +40%) for both the NNCLR baseline and ManifoldCLR. Second, for reusability of existing code, we do not include the additional prediction head used in NNCLR. This, however, has very minimal impact (~0.4%), as shown in Table 7f of NNCLR [1].

We hope these experiments help with any doubts about the generality of ManifoldCLR to point pair strategies. We are happy to answer any additional questions about the experimental setup or the results. We are still working to update our manuscript as soon as possible with these experiments and the other presentation details requested by reviewers, but hope that we have now addressed any technical concerns with the work.


|                             | Linear Separability |
|-----------------------------|------------------|
| NNCLR                       | 66.13 +/- 0.20   |
| ManifoldCLR (NN Point Pair) | 69.43 +/- 0.17   |


[1] “With a Little Help from My Friends: Nearest-Neighbor Contrastive Learning of Visual Representations,” Dwibedi et al., 2021.

[2] “Learning Identity-Preserving Transformations on Data Manifolds,” Connor et al., 2023.

---

### Author Response · Authors · 2023-12-10
**Revised Manuscript Posted**

Dear Reviewers,

We are happy to share a revised manuscript with all requested changes. Please let us know if you have any additional questions or concerns.

Sincerely,
Authors

---

### Decision · Action_Editor_t5s7 · 2024-01-25

**Recommendation:** Accept with minor revision

**Comment:**

Given that the two criteria (Claims & Evidence and Audience) are satisfied, I recommend acceptance with the following minor revisions:

- As Lie groups are relatively less well-known within the machine learning community, the authors are encouraged to expand the background on Lie group operators and possibly integrate relevant portions of it into the main paper.
- The NNCLR results should be integrated into the main paper.
- Please check the equations in section 2.2.
- Please ensure proper formatting of references.

In addition, see the comments above regarding claims about the broader applicability of the method and ensure that any claims in the revised version of the paper (i.e. outside Section 4.2.1) are similarly scoped.

**Audience:**

The simplicity of the approach and the relatively novel use of Lie group operators suggests that the method would be of sufficient interest to the community.

**Claims And Evidence:**

This paper introduces a method called ManifoldCLR that directly models the image manifold in self-supervised learning with Lie group operators and uses variational sparse coding to infer the coefficients for a given pair of images. Experiments demonstrate that ManifoldCLR improves classification performance in both self-supervised and semi-supervised settings. A sufficient number of ablation studies and qualitative experiments show the effectiveness of the design decisions. The computational requirements of the method are also evaluated.

One of the reviewers raised concerns about the applicability of the proposed method to other self-supervised learning approaches. The updated experiments on NNCLR lend credence to the idea that ManifoldCLR is applicable more generally to InfoNCE-based self-supervised learning approaches. In my opinion, the claims in the first paragraph of Section 4.2.1 are indeed adequately supported given the new experiments.